# Altered Distribution and Expression of Syndecan-1 and -4 as an Additional Hallmark in Psoriasis

**DOI:** 10.3390/ijms23126511

**Published:** 2022-06-10

**Authors:** Eleni Koliakou, Manthou Maria Eleni, Ioanna Koumentakou, Nikolaos Bikiaris, Polyanthi Konstantinidou, Patricia Rousselle, Doxakis Anestakis, Elisabeth Lazaridou, Evangelia Kalloniati, Dimosthenis Miliaras, Anna Michopoulou

**Affiliations:** 1Laboratory of Histology and Embryology, Medical School, Aristotle University of Thessaloniki, 54124 Thessaloniki, Greece; elenakoliakou@gmail.com (E.K.); mmanthou@auth.gr (M.M.E.); miliaras@auth.gr (D.M.); 2Laboratory of Polymer Chemistry and Technology, Department of Chemistry, Aristotle University of Thessaloniki, 54124 Thessaloniki, Greece; iwanna.koumentakou@gmail.com (I.K.); nbikiaris@gmail.com (N.B.); 3Department of Pathology, Forensic Service of Thessaloniki, 57003 Thessaloniki, Greece; polyanth@auth.gr (P.K.); anestaki@auth.gr (D.A.); 4Laboratoire de Biologie Tissulaire et Ingénierie Thérapeutique, UMR 5305, CNRS, University Lyon 1, SFR BioSciences, 7 Passage du Vercors, 69367 Lyon, France; patricia.rousselle@ibcp.fr; 52nd Department of Dermatology-Venereology, Medical School, Papageorgiou Hospital, Aristotle University of Thessaloniki, 54124 Thessaloniki, Greece; bethlaz@auth.gr (E.L.); evakalloniati@yahoo.gr (E.K.); 6Biohellenika Biotechnology Company, Leoforos Georgikis Scholis 65, 55535 Thessaloniki, Greece; 7Laboratory of Biological Chemistry, Medical School, Aristotle University of Thessaloniki, 54124 Thessaloniki, Greece

**Keywords:** biological factors, electrospinning, gelatin patch, fibronectin, psoriasis, syndecan-1, syndecan-4

## Abstract

Syndecans act as independent co-receptors to exert biological activities and their altered function is associated with many pathophysiological conditions. Here, syndecan-1 and -4 were examined in lesional skin of patients with psoriasis. Immunohistochemical staining confirmed altered syndecan-1 distribution and revealed absence of syndecan-4 expression in the epidermis. Fibronectin (FN)—known to influence inflammation and keratinocyte hyperproliferation via α5β1 integrin in psoriasis—was also decreased. Syndecan-1 and -4 expression was analyzed in freshly isolated lesional psoriatic human keratinocytes (PHK) characterized based on their proliferation and differentiation properties. mRNA levels of syndecan-1 were similar between healthy and PHK, while syndecan-4 was significantly decreased. Cell growth and release of the pro-inflammatory Tumor Necrosis Factor-alpha (TNFα) were selectively and significantly induced in PHKs plated on FN. Results from co-culture of healthy keratinocytes and psoriatic fibroblasts led to the speculation that at least one factor released by fibroblasts down-regulate syndecan-1 expression in PHK plated on FN. To assay if biological treatments for psoriasis target keratinocyte proliferation, gelatin-based patches enriched with inteleukin (IL)-17α or TNFα blockers were prepared and tested using a full-thickness healthy epidermal model (Phenion^®^). Immunohistochemistry analysis showed that both blockers impacted the localisation of syndecan-1 within the refined epidermis. These results provide evidence that syndecans expression are modified in psoriasis, suggesting that they may represent markers of interest in this pathology.

## 1. Introduction

Psoriasis vulgaris is a chronic, multifactorial skin disease belonging to the so-called immune-mediated disorders. Psoriasis manifestation involves the interaction of genotype, environmental, infectious, and lifestyle factors [1].

Psoriasis is characterized by keratinocyte hyperproliferation and aberrant differentiation, neoangiogenesis, and inflammation. In chronic plaque lesions, a marked epidermal thickening is observed [2,3], as a result of hyperplasia of basal and suprabasal keratinocytes accompanied by abnormal keratinocyte differentiation (expressed as altered keratins distributions) with marked parakeratosis (nuclei in the stratum corneum) and absence of granular layer [4,5,6].

Although psoriasis is clearly driven by T cell-mediated immune responses targeting the epidermis, intrinsic alterations in epidermal keratinocytes may also be involved. Keratinocytes of psoriatic skin are capable of initiating, sustain, and/or amplify the inflammatory responses by immune cells by being a source of growth factors (GFs) and cytokines expressed constitutively or after being triggered [7,8]. For example, the secretion of pro-inflammatory cytokines, such as IL-1β and TNFα, results in autoactivation of keratinocytes and production of more cytokines and chemokines [8,9]. In psoriasis, interferon-γ (IFN-γ), IL-17α, TNFα, and IL-22 can induce keratinocytes to produce more chemokines and cytokines in a self-amplifying loop [10]. Based on the findings of inflammatory cascades involved in the onset and maintenance of psoriasis, a wide range of biological therapies are currently licensed that target TNFα, IL-12/23, IL-17, and IL-17 receptor [11].

In parallel, alterations in the composition of the extracellular matrix (ECM), which is a source of directive contextual information to the cells, have already been reported to play an important role in the pathogenesis of the disease [12,13,14,15]. The best characterized change in the basement membrane (BM) of psoriatic skin is the over-expression of cellular fibronectin-EDA+ (cFN) and its receptor alpha 5 beta 1 (α5β1) integrin [16], as in neoplastic keratinocytes [17] and in keratinocytes involved in wound healing [13,18]. In psoriasis, FN is over-expressed in the papillary dermis where the laminin interruptions of the BM occur [19]. In the stratum corneum above the rete ridges, intraepidermal FN synthesis by keratinocytes and Langerhans cells is also increased [15]. Aberrant localization of the embryonic form of cFN is seen in proliferating, developing tissue and in wound healing and in nonlesional psoriatic skin in vivo [20]. It was recently shown that FN is abnormally regulated in psoriatic non-lesional skin and with integrin α5β1 influence inflammation and keratinocyte hyperproliferation [21].

Syndecans are attracting attention for their potential involvement in the development of inflammatory diseases [22], including psoriasis [23], rheumatoid and psoriatic arthiritis [24], inflammatory bowel disease [25], pemphigus vulgaris, bullous pemphigoid, and atopic dermatitis [26,27]. Syndecans were initially considered as co-receptors but there is now evidence supporting that they can independently exert biological activities. Syndecans are transmembrane proteoglycans constituted of a core protein onto which glycosaminoglycan (GAGs) chains are attached [28]. The cytoplasmic domain does not possess intrinsic kinase activity. Nevertheless, both the transmembrane and cytoplasmic domains contain highly conserved regions that are indicative of conserved interactions and functions [28,29]. Syndecans are known to interact with a large number of molecules, including adhesion receptors, cytokines, chemokines, GFs, proteinases, and other ECM proteins [30]. Moreover, it is known that under physiological conditions the ectodomains of syndecans are constitutively shed, thus existing as soluble forms [31]. More recently, a role in the regulation of cell cycling and gene transcription has been attributed to syndecans through mechanisms that involve their translocation into the nucleus [32]. Therefore, syndecans can be implicated in numerous different signaling events involving cell adhesion, cell-ECM interactions, migration, keratinocyte proliferation, and differentiation, inflammation, and wound healing [22,31,33,34].

To date, four syndecans have been identified, which are expressed in a cell-, tissue-, and development-specific manner. In mature tissue, epithelial cells permanently express syndecan-1, while syndecan-4 is expressed in a rather universal manner [25]. In the healthy skin, syndecan-1 is distributed pericellularly in all living epidermal layers, with stronger expression in the suprabasal region. Syndecan-4 shows pericellular localization in keratinocytes of the basal and spinous layers [33]. Indeed, there are a few studies correlating the expression levels and/or patterns of these two syndecans in different skin inflammatory diseases.

The roles of syndecans often seem to be controversial, depending on the tissues in which they are expressed and their physiological condition but also whether they are shed from the membrane or not [35]. One general observation is that in pathological conditions the total expression of syndecans is up-regulated, while the level of syndecans in the serum of animals can be correlated directly to the severity of inflammation [22]. However, syndecans can both exert pro- and anti-inflammatory roles, as shown in knockout models [22]. These differences observed in different inflammatory conditions suggest the potential of syndecans as bio-markers of several diseases.

In this study, we looked for evidence implicating the syndecan-1 and syndecan-4 expressed within the human epidermis with the pathogenesis of psoriasis. To do so, we used several approaches to evaluate the expression pattern of syndecans and propose a possible role in the pathogenesis of psoriasis involving immunohistochemistry analysis of skin biopsies, gene expression analysis, cell treatments, and application of biological factors onto skin equivalent (SE) models. For the latter, we used a gelatin nanofiber patch that we fabricated using the electrospinning technique based on previously published report [36]. The selection of the patch was inspired by reported related beneficial properties in supporting cell attachment and proliferation. Gelatin nanofiber patches have been used for a multitude of biomedical applications such as scaffolds used in tissue engineering [37,38,39], wound dressing [40], and drug delivery [41], presenting excellent results. Once produced, the patches were enhanced with two different -TNFα or IL-17α- blockers selected as commonly used biological factors routinely administered systematically for the treatment of psoriasis.

## 2. Results

### 2.1. Altered Expression and Distribution of Syndecan-1 and -4 and FN in Plaque Psoriasis

In healthy skin, syndecan-1 was expressed in almost full thickness of the epidermis. The strongest staining was observed in membranes and intercellular junctions of spinous and granular layers and the lowest in the cytoplasm of basal layer keratinocytes (Figure 1a. In psoriatic skin, syndecan-1 expression was rather strong in the membranes and intercellular junctions of cells of the basal layer and, specially, within the elongated rete ridges of the psoriatic epidermis. (Figure 1b,c). Syndecan-4 was expressed mainly in the periphery and cytoplasm of basal layer cells of healthy epidermis and at lower levels in the spinous layers (Figure 1D). In psoriatic epidermis syndecan-4 was absent from all the layers (Figure 1e,f). Cellular FN-EDA domain staining was observed within the cytoplasm of basal layer keratinocytes of non-psoriatic skin (Figure 1g). In the psoriatic epidermis no staining was observed intracellularly at any level. A very thin line of staining was occasionally observed below the basal epidermal layer of psoriatic skin likely corresponding to the basement membrane zone (Figure 1h,i). A very low staining of the 10th FN III (FNIII-10) domain was observed intracellularly in the basal layer of non-psoriatic skin (Figure 1J). Absence of staining was observed in the psoriatic epidermis. Some non-specific staining observed at the upper layers may be due to the presence of plasma FN after hemorrhage at the region of the lesion (Figure 1k,l). The quantification of each respective staining by epidermal layer is presented in Table 1.

### 2.2. Freshly Isolated Keratinocytes from Psoriatic Lesions Cultured In Vitro Present Differences Insyndecan-1 and -4 Expression at the Gene Transcription Level While Maintain an “Active” Phenotype

Keratinocytes within the psoriatic lesion are found at an “active” state characterized in short by excessive proliferation, abnormal differentiation, and chemokine/cytokine expression. Interestingly, as shown in Figure 2a, when early passage psoriatic keratinocytes (passage 0 to 1) were cultured for 24 h onto plastic presented a significantly lower growth or viability rate than healthy keratinocytes, as revealed by MTT analysis. MTT is a semi-quantitative assay that determines cell proliferation via the estimation of the metabolic activity of cells in cell culture. MTT is relatively straightforward and easy to use but has some limitations among which is the effect of cell seeding density [42]. MTT was used to evaluate and compare the proliferation rates of healthy versus psoriatic keratinocytes. Cultured psoriatic keratinocytes observed to the light microscope at different time points (data not shown here) appeared to behave differently than healthy keratinocytes. Usually, keratinocytes in cell culture grow actively and start to differentiate when they are in contact with other cells which is also an indication of the culture confluency. Nevertheless, we made the observation that psoriatic keratinocytes in cell culture presented alternative cycles that the cells were sporadically more or less confluent independent on the day of culture instead of increasing progressively. It is likely that this this “phenomenon” reflects the deregulated processes of cell proliferation and differentiation in psoriatic keratinocytes, and which might have interfered with the data collected by MTT analysis. Figure 2b, shows K6, K10 and K14 expression patterns in healthy and psoriatic keratinocytes. All the three keratins are expressed at higher levels in the psoriatic keratinocytes in cell culture in vitro, while expression of K6 was significantly higher. Interestingly, gene expression analysis of syndecan-1 and -4 were likely confirmatory of the findings from the histological analysis. As shown in Figure 2c, while syndecan-1 expression was not significantly different between the two groups syndecan-4 was expressed significantly less within the psoriatic keratinocytes.

### 2.3. Psoriatic Keratinocytes Respond Differently Than Healthy Keratinocytes upon Attachment to FN

The aim of this experiment was to assay the response of psoriatic keratinocytes upon attachment to a recombinant FN concerning two important aspects in the pathogenesis of psoriasis which is cell growth and cytokine release. As shown in Figure 3a, the attachment to FN resulted in a significantly higher growth of psoriatic keratinocytes as compared to the healthy. This treatment also induced a significant increase in the release of TNFα in the cell culture supernatant of psoriatic keratinocytes (Figure 3b). More interestingly, when healthy keratinocytes were co-cultured with psoriatic fibroblasts there was an increase in syndecan-1 expression which was significantly reversed when FN was added as a substrate for keratinocytes in the same culture system. Syndecan-4 expression presented no alterations in this experimental setting (Figure 3c).

### 2.4. Characterization of Gelatin Patch

The possibility of structural change of gelatin molecules after of dissolution in acetic acid and electrospinning process was examined by FT-IR spectroscopy. As shown in Figure 4, gelatin powder and gelatin patch illustrated common bands of protein presenting the same characteristic peaks at 1230 cm^−1^ due to C-N, N-H vibrations in amide III whereas 1534 cm^−1^, 1660 cm^−1^, and 3304 cm^−1^ are the peaks corresponding to N-H and C-N vibrations in amide II, to C=O and N-H vibrations in amide I, and O–H stretching vibration, respectively. Therefore, FTIR spectra confirmed that acetic acid and electrospinning process did not affect the chemical structure of gel [43].

In order to examine the effect of electrospun nanofiber on gelatin crystal structure, XRD patterns were investigated. As shown in Figure 4, gelatin powder gave a typical XRD pattern of gelatin hemicrystalline structure originated from α-helix and triple-helical structure. However, the electrospun patch prepared from gelatin solution showed less developed crystal structure and lower crystallinity. The crystalline structure of gelatin seemed to be destroyed by dissolving the gelatin in acetic acid solution, and therefore, the gelatin nanofiber web becomes mostly amorphous [44].

The surface morphology of gelatin patch was investigated by SEM micrographs and ImageJ analysis. Concerning the fiber diameter and pore sizes of the prepared patch, it was observed that the surface presented a well-defined structure. As long as the gelatin solutions were spinnable, the resulting fibers presented diameters ranging from 209 ± 0.26 to 450 ± 0.23 nm. The prepared nanofiber network showed a random arrangement, forming pores whose dimensions varied from 0.36 ± 0.16 to 0.82 ± 0.9 µm for pore widths and 0.45 ± 0.32 to 0.98 ± 1.66 µm for pore lengths ranged.

### 2.5. Data from a Reconstructed Skin Model Showing That Treatment with a Gelatin Patch as Carrier of TNFa and IL-17a Blockers Affect Epidermal Syndecan-1 but Not Syndecan-4 Expression

Firstly, the impact of the patches onto epidermal thickness in the reconstructed skin models was assayed as shown in Figure 5a,b. Statistical analysis demonstrated that both anti-TNFα- and anti-IL-17α-enriched patches resulted in a significant decrease in epidermal thickness. MTT analysis showed no significant difference in proliferation rates between the different patches and the control suggesting that the new devices had no toxic effect to the skin or that the patches do not affect cell proliferation (Figure 5c). However, this assay was performed on full-thickness SEs involving both keratinocytes and fibroblasts. To further rule out a possible contact sensitization effect of the patches we performed ELISA for IL-18 detection in the SEs culture medium (Figure 5d). To more specifically look at the effect of the treatments on the epidermal activity we performed immunohistochemical analysis for PCNA proliferation marker, K14 basal keratinocytes marker and K6 activated keratinocytes marker (Appendix A). Our results demonstrated a significant decrease of both K14+ and K6+ keratinocytes (Appendix A) in the healthy SE model after treatment with TNFα blocker. Further evaluation using a psoriasis SE model will be necessary to exclude the possibility of a deleterious effect of anti-TNFα treatment to the epidermis. Finally, immunohistochemical staining (Figure 5d) revealed an important impact on syndecan-1 detection after treatment with anti-TNFα- and anti-IL-17α-patches, with some areas where syndecan-1 was almost totally absent from the epidermis. Syndecan-4 staining remained unaltered.

## 3. Discussion

In this paper, we used several approaches to assess the potential significance of epidermal syndecans’ expression and distribution in the pathogenesis of psoriasis. It has been previously shown that the pattern of expression of syndecan-1 is altered in the psoriatic epidermis with the expression to be diminished with cell differentiation [31]. This study confirmed the alteration in syndecan-1 distribution within the psoriatic lesion and showed for first time the absence of syndecan-4 (Figure 2d).

To study the expression levels of syndecan-1 and -4 in the lesional epidermis we used freshly isolated keratinocytes. We firstly tried to confirm that psoriatic keratinocytes maintain their phenotype during cell culture. In Figure 3a and b we show that culturing psoriatic keratinocytes onto FN induced an up-regulated cell proliferation and significant pro-inflammatory TNFα release, which in accordance with the hypothesis that keratinocytes in psoriasis are in a pre-activated state and respond differently to FN than healthy keratinocytes. Our RT-PCR analysis showed a non-significant decrease of syndecan-1 and an important decrease of syndecan-4 at the gene transcription level in freshly isolated keratinocytes from the psoriatic lesion (Figure 1b). Interestingly, an increase of syndecan-4, was previously observed in the serum and lesional skin of patients with atopic dermatitis [27] suggesting that the presence of this molecule might differentiate between the two skin diseases.

As mentioned before, a well-characterized alteration in the ECM in psoriasis is the presence of FN in the BM accompanied by a redistribution of its receptor integrin α5β1 in the basal cell layer of the epidermis of psoriatic uninvolved skin [16,20,21]. In this study, the detection of both pFN and cFN was not significant in the histological sections of lesional skin (Figure 1a) implying that its presence is specific within the uninvolved skin (not examined in this study). Moreover, keratinocytes from healthy skin in co-culture with psoriatic fibroblasts only down-regulated syndecan-1 expression when they were grown onto a FN substrate (Figure 3c). This result suggests that psoriatic fibroblasts impact the keratinocytes through their interaction with FN. This result is not the first one showing the regulation of syndecan-1 expression by ECM components. It was demonstrated before that syndecan-1 expression is activated in keratinocytes cultured of FN upon treatment with keratinocyte growth factor (KGF) through a mechanism involving the fibroblast growth factor-inducible response element (FiRE) [45]. In our experimental conditions, syndecan-4 expression was not affected suggesting that each syndecan may have distinct roles in the pathogenesis of the disease. Syndecan-1 expression in epidermal keratinocytes is inducible and depends on events associated with differentiation or repair of injuries [46]. A possibility might be that syndecan-1 is down-regulated in the keratinocytes of non-lesional skin in response to exposure to FN in combination with some GF released by psoriatic fibroblasts, while it re-arranges in the lesional epidermis. It has been previously shown that cell adhesion to FN requires two independent signaling events involving the cell-binding domain and the heparin-binding domain of the molecule that are mediated by transmembrane receptors including integrins and the superfamily of cell-surface proteoglycans syndecans. Previous and recent data suggest that two members of the family, syndecan-1 and -4 are related to signaling downstream to FN exposure [47]. Syndecan-4 is known to play a role in the formation of focal adhesions in co-operation with integrin α5β1 [48,49,50] while syndecan-1 was shown to activate integrins and regulate focal adhesion assembly and FN fibrillogenesis during corneal cell migration in the context of wound healing [51]. Consequently, we can speculate that syndecans may be implicated in cell signaling or be affected by FN in psoriasis, too, at least for syndecan-1 according to our data. Another hypothesis is that both syndecans are likely to have a role to play in psoriasis, but their contribution is spatially and temporally differentiated.

To enhance our data supporting the participation of syndecans in the molecular mechanisms that lead in psoriasis, we tested their potential as candidate targets for treatment. To do so, we used the Phenion full-thickness reconstructed skin model to apply epidermally two different biological factors already used to treat patients with psoriasis, namely, anti-TNFα and anti-IL-17α. However, instead of applying the biologics directly onto the epidermis we prepared a gelatin-based scaffold for the delivery. The idea of doing that was to propose at the same time an alternative method for the administration of these molecules that are routinely administered intravenously with many undesirable side effects for the patients [52]. Gelatin patches have already been assayed for the treatment of wound, they are relatively easy to prepare biodegradable, biocompatible, and cost-effective [53,54]. Additionally, previous works proved that gelatin patch with nanofibers could be easily used to achieve encapsulation of biologics or other factors/drugs for their controlled delivery and penetration to the epidermis [41,55,56,57]. The current work demonstrated (Figure 4) that the fabricated gelatin patch presented suitable morphology (the size of nanofiber and porous structure) to support the topical release of anti-TNFα and anti-IL-17α suggesting future developments to adapt their application for the treatment of psoriasis. Furthermore, as shown in MTT test (Figure 4), the gelatin patch presented excellent biocompatibility. The patch was also evaluated after being enhanced with the blockers as shown in Figure 5a–c. In fact, applying both anti-TNFα and anti-IL-17α resulted in a significant decrease in epidermal thickness which in the context of psoriasis would be a desirable effect. However, we used a healthy SE model and in order to verify if this decrease is due to cytotoxic effects of the patches, we performed a MTT analysis which excluded this possibility. By analyzing IL-18 release we could also demonstrate that there is not a contact sensitivity reaction caused by the application of the patches neither. Finally, the application of the patches containing both TNFα and IL-17α blockers had an effect on syndecan-1 distribution within the epidermis, while syndecan-4 was not affected (Figure 5d), suggesting that syndecan-1 could be a valuable marker. Whether this is a positive outcome for the treatment of the disease requires thorough examination. It is now necessary to perform these experiments using a psoriasis skin model to find out whether these observations are consistent. Moreover, other factors or drug that would specifically target syndecans or syndecan-related molecular pathways will be interesting to be investigated in the future.

## 4. Materials and Methods

### 4.1. Patients and Samples

Lesional skin samples were collected from the elbow or knee of 7 patients (4 men and 3 women) with chronic psoriasis vulgaris, aged 36–66 y old. An equal number of healthy volunteers, aged 37–70, donated skin samples removed after their scheduled surgery. All patients had not been treated for at least 1 m by topical and/or systemic treatment at the time that the biopsy was taken. Equal number of biopsies was collected from healthy subjects. 6 mm punch biopsies were obtained after local anesthesia with 1% buffered xylocaine. All samples were obtained after informed consent, according to the Declaration of Helsinki and the medical ethical committees of Aristotle’s University of Thessaloniki and General Hospital Papageorgiou.

### 4.2. Monolayer Cell Cultures and Co-Cultures

Human epidermal keratinocytes and fibroblasts cultures were established from skin specimens according to previously published procedures [58]. Keratinocytes were grown in keratinocyte growth medium (DermaLife K basal medium, (# LL-004, CELLSYSTEMS, Troisdorf, Germany) supplemented with L-Glutamine, apo-transferrin, recombinant human (rh) transforming growth factor alpha (TGF-α), rh insulin, hydrocortisone, epinephrine, bovine pituitary extract-BPE, and gentamycin/amphotericin included in the kit (DermaLife K LifeFactors kit, (# LS-1030, CELLSYSTEMS, Troisdorf, Germany). Fibroblasts were grown in DMEM w/L-glutamine (#L0104-500, BIOWEST, Nuaillé, France) supplemented with 10% fetal bovine serum (FBS) and penicillin-streptomycin (BIOWEST, # L0018-100). For co-cultures, 50,000 psoriatic fibroblasts were cultured per well of 12-well plates (Greiner bio-one, #665-180-01, München, Germany) in DMEM w/L-glutamine (BIOWEST, #L0104-500) supplemented with 10% fetal bovine serum (FBS) and penicillin-streptomycin (# L0018-100, BIOWEST, Nuaillé, France). For the co-culture experiments fibroblasts were cultured in the wells of a 12-well plate for 2 w in DMEM culture medium with FBS and 50 ug/mL ascorbic acid. Then, 50,000 keratinocytes were seeded on top of fibroblasts in the upper chamber of the cell culture insert (Thincert, 0.4um pore diameter, #665640, Greiner bio-one, München, Germany) for 24 h and then cell lysates were collected for further analysis.

### 4.3. Cell Treatments

For induction experiments, recombinant fibronectin (# FC010, Hu Plasma Fibronectin, MILLIPORE, Berlington, Massachusetts, USA) at a concentration of 20 μg/mL was plated onto the bottom of 24-well culture plates and cells at 50.000 per well were left to attach. When human keratinocytes from 3 healthy and 3 patients with psoriasis attached to the plates, the complete DermaLife K medium was replaced by medium without additives for 24 h. Cell supernatants were collected for analysis of their TNFα content by ELISA. In parallel, the cells were subjected to MTT test to analyze the effect of fibronectin onto keratinocytes’ proliferation. For the co-cultures, the inserts were pre-coated with 20 μg/mL fibronectin before seeding keratinocytes and cultured for 24 h. As controls, keratinocytes onto plastic and co-culture of psoriatic fibroblasts with keratinocytes cultured on inserts without coating were used. The experiment was performed in triplicates.

### 4.4. Immunohistochemistry

The skin biopsies were fixed in 4% paraformaldehyde (PFA) solution for 24 h at 4 °C, then dehydrated in ascending alcohol concentrations and embedded in paraffin. Millipore’s IHC Select^®^ HRP/DAB detection kit (DAB150) and anti-Syndecan-1 (A-6) (# sc-390791, Santa-Cruz Biotechnology, Inc., Dallas, TX, USA), anti-Syndecan-4 (5G9) (# sc-12766, Santa-Cruz Biotechnology, Inc., Dallas, TX, USA), anti-plasma-Fibronectin (C6F10) (# sc-73611, Santa-Cruz Biotechnology, Inc., Dallas, TX, USA), anti-ED-A-cellular-Fibronectin (IST-9) (# sc-59826, Santa-Cruz Biotechnology, Inc., Dallas, TX, USA),), anti-keratin (K)-6 (clone W170068, isotype rat IgG2a, # 606101, Biolegend, San Diego, CA, USA), anti-K14 (clone poly19054, isotype rabbit polyclonal IgG, # 905403, Biolegend, San Diego, CA, USA), anti- Proliferating Cell Nuclear Antigen (PCNA) (clone PC10, isotype mouse IgG2a, # 307901, Biologend. San Diego, CA, USA), and secondary antibodies Goat anti-rabbit IgG-HRP (# sc-2004, Santa-Cruz Biotechnology, Inc., Dallas, TX, USA), Goat anti-mouse IgG-HRP (# sc-2031, Santa-Cruz Biotechnology, Inc., Dallas, TX, USA) were used for immunohistochemical staining of the biopsies. Deparaffinization was performed on PT module system, with pH retrieval high. Slides then incubated in PBS for 10 min, proteins were blocked for 5 min by the Blocking Reagent, primary antibodies were added for 30 min (diluted 1: 200 in PBS), then slides incubated for 10 min at secondary reagent (biotinylated goat anti-mouse IgG and goat anti-rabbit IgG in PBS, containing carrier protein), and, finally, 10 min incubation in Streptavidin HRP. Section staining was completed by adding DAB dye for 1 min, washing with distilled water and then adding hematoxylin dye for 2 min. The sections were immersed in 96% alcohol for 2 min, repeated 2 times, and xylene baths alike, and covered with cover strips.

### 4.5. Enzyme-Linked Immunosorbent Assay (ELISA)

Healthy and psoriatic keratinocytes attached to recombinant FN, as described above, were assayed for TNFα release by examination of the cell culture supernatant using an ELISA kit (#ΕΚ-072-28, Phoenix pharmaceuticals, Inc., Burlingame, CA, USA) according to the manufacturer’s instructions. An ELISA assay was also performed for the detection of IL-18 in the culture media of SEs after exposure to biological factors-containing patch using a commercial kit according to the manufacturer’s instructions (# CSB-E074504, Fannin St Ste 200-V126, Houston, USA).

### 4.6. MTT (3-[4,5-Dimethylthiazole-2-yl]-2,5-diphenyltetrazolium Bromide) Proliferation Assay

MTT assays were performed in 24-well plates at cell densities of 50,000 cells/well and at passage 0 or 1 for the experiments performed with keratinocytes. MTT was used for the comparison of proliferation rates between human keratinocytes from healthy and psoriatic keratinocytes from lesional skin and also for the FN attachment experiments. Furthermore, MTT was performed to assay biocompatibility of the gelatin patch using human fibroblasts seeded at 5000 cells/well into 96-well plates. MTT was also performed on SEs for irritation testing. MTT was used at a concentration of 0.5mg/mL. Then, incubation at 37 °C, CO_2_ 5% for 4 h was performed until intracellular purple formazan crystals were visible under microscope. MTT was removed and solubilizing DMSO solution was added for 30 min, until cells were lysed, and purple crystals were resolved. Then, the supernatants were transferred to a new plate for reading the optical density at the spectrophotometer at 570/630 nm.

### 4.7. Real-Time Reverse Transcription PCR

Total RNA was isolated from monolayers using a Nucleospin RNA II kit (MACHEREY-NAGEL EURL, Hoerdt, France) according to the manufacturer’s instructions. 100 ng of total RNA extracted from healthy or psoriatic keratinocytes (until passage 1) were subjected to the first strand cDNA synthesis and QPCR analysis using the KAPA SYBR FAST One-Step qRT-PCR KIT (Kapa Biosystems, Wilmington, MA, USA). The primers (Eurofins Genomics AT GmbH) used for the detection of gene expression are listed below. Reactions were performed using a Corbett 6000 real-time PCR cycler Rotorgene. The presence of a single dissociation peak was verified by melt curve analysis. Relative quantification was determined using the comparative (Ct) method [59] with normalization to the housekeeping gene GAPDH.

Syndecan-1 FP: CTTCACACTCCCCACACAGA, RP: GTATTCTCCCCCGAGGTTTC, Syndecan-4 FP: TGAGGATGTGTCCAACAAGG, RP: AGGAAGACGGCAAAGAGGAT K6: FP: GTGAGGAGTGCAGGCTGAAT, RP: CATAGCCACTGGAGACGGTG, K14: FP: TCATCCAGAGATGTGACCTCC, RP: GCCTCAGTTCTTGGTGCGAA, K10: FP: TGATGTGAATGTGGAAATGAATGC, RP: GTAGTCAGTTCCTTGCTCTTTTCA, GAPDH: FP: TGCACCACCAACTGCTTAGC, and RP: GGCATGGACTGTGGTCATGAG.

### 4.8. SE Model

The SE full-thickness model was purchased (Phenion FT Skin Model, # FT SM-1) and air-liquid interface (ALI) medium (#ALI CM-250) were purchased by Phenion Henkel AG and Co, Düsseldorf, Germany. These in vitro models consist of normal human-derived epidermal keratinocytes and fibroblasts cultured at the ALI to induce epidermal formation and differentiation. The SE were handled according to manufacturer’s guidelines and were cultured at 37 °C, CO_2_ 5% for another 5 d since their arrival in ALI medium before use.

### 4.9. Synthesis and Characterization of Gelatin Electrospun Patch

Gelatin solution was produced to fabricate the nanofiber patch. Specifically, 20% *w*/*v* gelatin (from bovine skin, Sigma-Aldrich, St. Louis, MO, USA) was dissolved in acetic acid (purity 99.8%, Sigma-Aldrich, St. Louis, MO, USA) solution 80% *v*/*v* stirring at 60 °C for 4 h to promote a complete dissolution of the polymer.

The polymer patch was fabricated, using the electrospinning apparatus (FLUIDNATEK^®^ LE-10, Bioinicia, Valencia, Spain), equipped with a 10 cm diameter rotating drum collector covered by an aluminum foil. 10mL of gel solution was poured into a syringe pump to prepare the electrospun nanofiber patch. The standard parameters used to electrospun gelatin dissolved in acetic solution were: voltage 20 kV, needle-collector distance 10.0 cm, flow rate 2000 μL/h, and needle 20 G. The prepared electrospun patch was dried for 24 h at room temperature to remove the presence of solvent.

The chemical structure of the synthesized patch was determined by FT-IR spectroscopy. FT-IR spectra were obtained using a FTIR-2000 (Perkin Elmer, Dresden, Germany) with KBr discs. The spectra were collected in the range from 4000 to 400 cm^−1^ at a resolution of 4 cm^−1^ using 16 coadded scans while the baseline was corrected and converted into absorbance mode. The crystal structure of patch was studied by X-ray diffraction (XRD) patterns, which were performed with an XRD-diffractometer (Rigaku Miniflex II, Beijing, China) with CuKα radiation for crystalline phase identification (λ = 0.15405 nm for CuKα). The sample was scanned from 5 to 60, with steps of 0.05°. The prepared nanofibers of patch were investigated by scanning electron microscopy (SEM) images, which were obtained with an electron microscope JEOL 2011 (Akishima, Tokyo, Japan). Each studied sample was placed on the holder and covered with carbon to provide good conductivity of the electron beam. Operating conditions were set at accelerating voltage 20 kV, probe current 45 nA and counting time 60 s. ImageJ software was used to analyses the pore size and fiber diameter from the top surface of patch and all measurements were obtained in triplicate.

### 4.10. Testing of the Anti-TNFa or Anti-IL-17A Enriched Patches onto the SEs

The TNFα blocker used for this study was a monoclonal human anti-TNFα antibody (about 234 amino acids) produced in mouse (clone M1-C4, # SAB1404480, Sigma-Aldrich, St. Louis, MO, USA). The antibody against human IL-17α was produced in rabbit (about 18kDa) (# DF6127, Affinity Biosciences). The collagenous patches were soaked for 5 min either in an anti-TNFα solution or an anti-IL-17α solution of a concentration of 1 μg/mL and was placed against the epidermal surface of the SE for 35 min. Then, the SEs were either analyzed by MTT for skin viability assay or subjected to histological analysis. The cell culture media were collected for the determination of contact sensitization potential through measurement of IL-18 by ELISA.

### 4.11. Image Analysis

Semi-quantitative analysis was performed to analyze staining intensities in histological sections. Basic analysis involved direct scoring by eyes under microscope as follows: negative (−), weak (+), moderate (++), and strong (+++). Epidermal thickness was measured as the average of 3 measurements made along the length of 3 representative H and E-stained histological sections per experimental condition using ImageJ.

### 4.12. Statistical Analysis

All the graphs, calculations and statistical analyses were performed using GraphPad Prism software version 9.3.1 (GraphPad Software, San Diego, CA, USA). Data were presented as mean ± SD and Student’s t-test (unpaired, two-tailed) was used for two-group comparisons. The exact sample size (n) for replicate measurements of specific keratinocyte strains or of groups of individual donors are specified in each graph legend. Differences were considered statistically significant at a value of *p* < 0.05. For more than two-group comparisons multiple *t*-tests were determined in GraphPad Prism v. 9.3.1. by analyzing each raw individually without assumption of consistent standard deviations, unpaired *t*-test with Welch correction (Two-stage step-up method of Bonjamini, Krieger and Yekutieli). Differences were considered statistically significant at a value of *p* < 0.05.

## 5. Conclusions

Our study provides evidence for the involvement of syndecan-1 and/or syndecan-4 in the pathogenesis of psoriasis and could serve as potential targets for treatment. It was confirmed that gelatin patch with nanofibers could be used as a carrier topical delivery of biological agents for potential psoriasis treatment. However, further research should be done on the stability and mechanical properties of patch which could be reinforced with other synthetic or natural polymers [37,52,56,57]. Moreover, further study should be done to confirm the benefit of applying biologics topically instead of systemically for the treatment of psoriasis and gelatin patches appear to have good potential to that end.

## Figures and Tables

**Figure 1 ijms-23-06511-f001:**
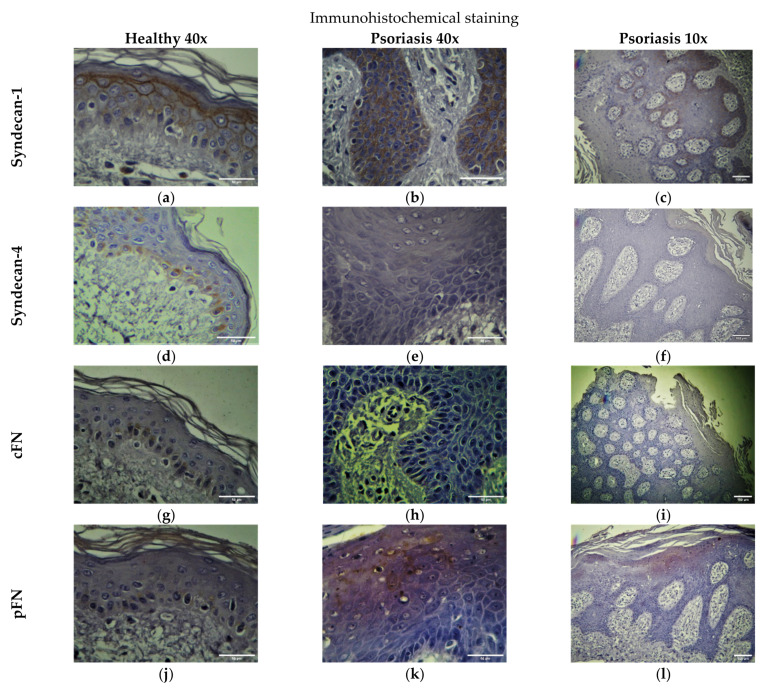
Immunohistochemical analysis of skin biopsies from lesional skin of patients with psoriasis. (**a**) Syndecan-1 expression in non-psoriatic, healthy epidermis. (**b**,**c**). Syndecan-1 expression was rather strong in the psoriatic epidermis at magnifications 10x and 40x. (**d**) Syndecan-4 in non-psoriatic epidermis. (**e**,**f**). Expression of syndecan-4 in psoriatic epidermis at magnifications 10X and 40x. (**g**) Cellular FN-EDA domain staining in non-psoriatic skin. (**h**,**i**). Patches of staining observed below the basal epidermal layer of psoriatic skin likely corresponding to the basement membrane zone at two magnifications. (**j**) Plasma fibronectin staining in the basal layer of non-psoriatic skin. (**k**,**l**). Plasma fibronectin staining in the psoriatic epidermis. (magnifications 10x, scale bar 100 μm, 40x, scale bar 50 μm).

**Figure 2 ijms-23-06511-f002:**
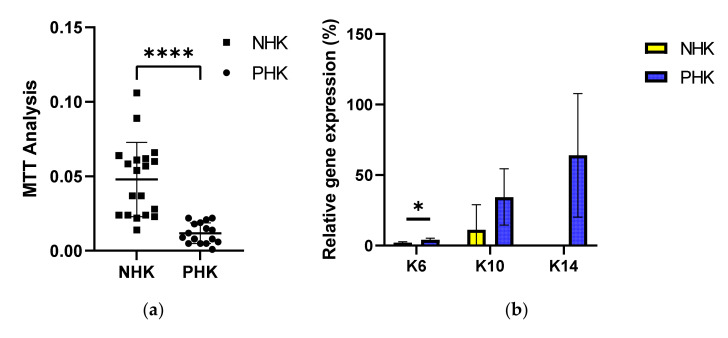
(**a**) Proliferation of normal human (NHK) and psoriatic keratinocytes (PHK) over 24 h. Four donors and five replicates per condition. Proliferation determined by MTT analysis. (**b**) Quantitative Real-Time PCR (QPCR) analysis of keratins (K6, K10, and K14) gene expression normalized to GAPDH in keratinocytes from lesional skin from 3 psoriatic donors vs. 3 healthy donors, plated on culture dishes. (**c**) QPCR analysis of syndecan-1 and -4 gene expression normalized to GAPDH in keratinocytes from lesional skin from 3 psoriatic donors vs. 3 healthy donors, plated on culture dishes. Experiments c and d were done in duplicates per donor, Mean ± (standard deviation) SD, * *p* < 0.05, **** *p* < 0.0001.

**Figure 3 ijms-23-06511-f003:**
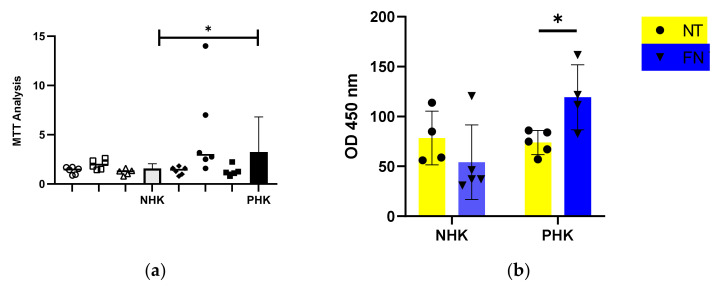
Keratinocytes from the psoriatic lesion respond differently upon attachment to a fibronectin (FN) substrate than healthy (**a**) MTT analysis of growth rate after 24 h attachment to a recombinant FN substrate. Three donors of each group were analyzed. The number of replicates per donor are demonstrated on the graph, mean ± SD, t-student test, * *p* < 0.05. (**b**) ELISA detection of TNFa in the cell culture medium in response to attachment to FN. One donor of each group was analyzed in such replicates as demonstrated on the graph mean ± SD, t-student test, * *p* < 0.05. (**c**) QPCR analysis of syndecan-1 and -4 expression in normal keratinocytes after co-culture with psoriatic fibroblasts with or without FN substrate. Experiment performed in triplicates, statistical analysis mean ± SD, multiple t-student tests, * *p* < 0.05.

**Figure 4 ijms-23-06511-f004:**
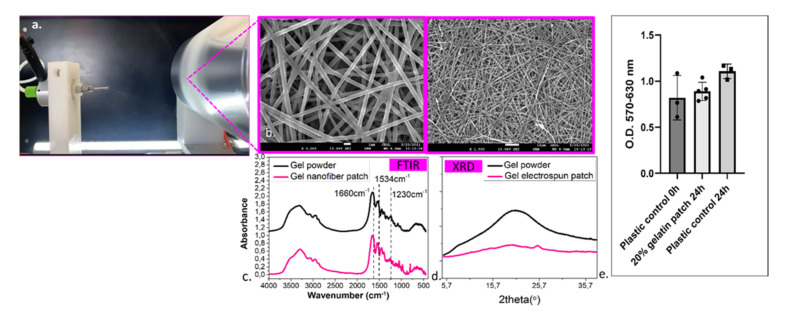
(**a**) Fabrication of gel patch with electrospinning process, characterization of gel patch, (**b**) SEM images, (**c**) FTIR spectra, (**d**) XRD patterns, and (**e**) MTT biocompatibility assay. The number of replicates per condition is indicated on the graph, statistical analysis mean ± SD, t-student test.

**Figure 5 ijms-23-06511-f005:**
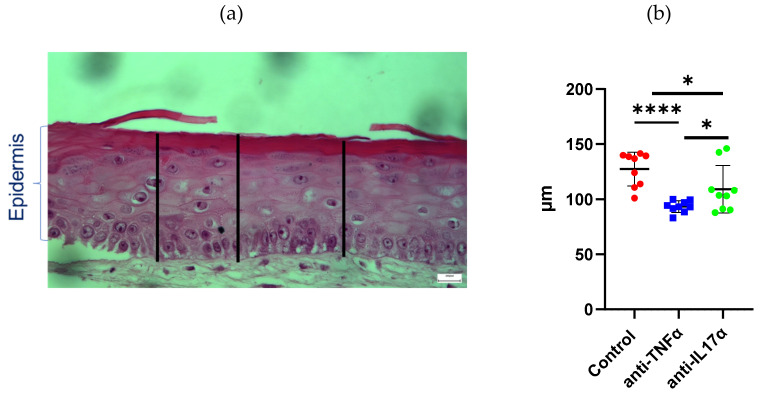
Effects of anti-TNFα and anti-IL-17α topical treatments onto the Phenion full-thickness skin model. H and E and immunohistological analysis of TNFα and IL17α blockers—treated SEs demonstrate reduction of epidermal thickness, accelerated differentiation process and altered syndecan-1 and -4 expression/distribution. (**a**) Epidermal thickness was assessed at regular intervals across the sections using ImageJ. (**b**) Mean epidermal thickness in microns. Three images corresponding to three independent experiments were analyzed per condition. Mean ± SD, t-student test for the comparison of two groups or multiple t-tests for more, * *p* < 0.05, **** *p* < 0.0001. (**c**) The biocompatibility was assayed using MTT proliferation analysis against control which was application of gelatin patch without blockers. (**d**) Contact sensitivity was assayed by analyzing IL-18 quantity (pg/mL) in the culture supernatant by ELISA at the same experimental conditions. (**e**) Evaluation of biological factors—enhanced gelatin patches using full-thickness reconstructed skin models. Immunohistological detection of syndecan-1 and -4 in reconstructed skin models after application of gelatin patches containing anti-TNFα or anti-IL17α.

**Table 1 ijms-23-06511-t001:** Quantification of staining in the histological sections.

	Syndecan-1	Syndecan-4	Plasma Fibronectin (FN-III-10)	Cellular Fibronectin (FN-ED-A)
Healthy skin				
Basal layer	−	++	+	+
Spinous layer	+	+	−	−
Granular layer	++	−	−	−
Psoriasis lesional skin				
Basal layer	+	−	−	−
Spinous layer	−	−	−	−
Granular layer	−	−	−	−
	Syndecan-1	Syndecan-4	Plasma Fibronectin (FN-III-10)	Cellular Fibronectin (FN-ED-A)
Healthy skin				
Basal layer	−	++	+	+

## Data Availability

Not applicable.

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
