# Peer review of "Altered Distribution and Expression of Syndecan-1 and -4 as an Additional Hallmark in Psoriasis"

_ijms, 2022, doi:10.3390/ijms23126511_

Round 1

Reviewer 1 Report

Review of manuscript ijms-1734653 entitled: Altered distribution and expression of Syndecan-1 and -4 as an additional hallmark in psoriasis and target for treatment   by Eleni Koliakou , Maria Eleni Manthou , Ioanna Koumentakou , Nikolaos Bikiaris , Polyanthi Papakonstantinou , Patricia Rousselle , Doxakis Anestakis , Elisabeth Lazaridou , Evangelia Kalloniati , Dimosthenis Miliaras , Anna Michopoulou

The manuscript is devoted to explanation of the some components of extracellular matrix role in development of psoriatic changes. Initially the altered expression of syndecan 1 was confirmed in epidermis. Decreased viability/proliferation of PHK was observed by MTT test. The results were not confirmed by PCNA expression. K6, K10 and K14 expression was confirmed in keratinocytes but higher level of K6 in PKH was observed. Lover level of syndecan-4 was expressed in PHK cells comparing with NHK. Higher growth of psoriatic keratinocytes attached to FN comparing to healthy cells was observed. Augmented TNFa concentration in medium of psoriatic keratinocytes was found. However in NHK/PHK/FN group expression of syndecan-1 was lower comparing with NHK/PHK group. Inhibitors of TNFa and IL7a altered syndecans distribution and reduced epidermal thickness

The manuscript could be interesting for the readers of International Journal of Molecular Sciences.

There are a several concerns with the study:

Major concerns

  1. The manuscript is not written clearly, thus it is difficult to understand the main idea.
  2. Why the Authors decided to check fibronectin effect on syndecans or to mix NHK with PHK in experiment showed on fig.3c. Moreover, rationale for inhibition of TNFa nad IL7a is scarce.
  3. Why the authors applied MTT test for proliferation of cells analysis. The results are not reliable. Even in the present paper the results obtained with MTT were inconsistent with PCNA expression. The better method is BrdU or other tests.
  4. Conclusion in the abstract suggesting that syndecans play a role in psoriasis and could be the target for therapy is not supported by the data.
  5. Mechanistic aspect of this study is scarce.
  6. The anatomical localization of the samples collection is not given.

Minor concerns

  1. The typing mistakes should be corrected

Author Response

Response to Reviewer 1 Comments

Point 1: The manuscript is devoted to explanation of the some components of extracellular matrix role in development of psoriatic changes. Initially the altered expression of syndecan 1 was confirmed in epidermis. Decreased viability/proliferation of PHK was observed by MTT test. The results were not confirmed by PCNA expression. K6, K10 and K14 expression was confirmed in keratinocytes but higher level of K6 in PKH was observed. Lover level of syndecan-4 was expressed in PHK cells comparing with NHK. Higher growth of psoriatic keratinocytes attached to FN comparing to healthy cells was observed. Augmented TNFa concentration in medium of psoriatic keratinocytes was found. However in NHK/PHK/FN group expression of syndecan-1 was lower comparing with NHK/PHK group. Inhibitors of TNFa and IL7a altered syndecans distribution and reduced epidermal thickness

The manuscript could be interesting for the readers of International Journal of Molecular Sciences.

There are a several concerns with the study:

Response 1: We thank the reviewer for taking the time to read and evaluate our paper. We addressed all comments and revised the manuscript accordingly.

Point 2: The manuscript is not written clearly, thus it is difficult to understand the main idea.

Response 1:

As mentioned in the discussion section of this mansucript, we used several approaches to assess the potential significance of epidermal syndecans’ expression and distribution in the pathogenesis of psoriasis. Our focus was to show/unveil possible alterations of syndecan-1 and/or syndecan-4 expression using different experimental conditions. Firstly, we showed the differences in expression and distribution in the skin biopsies. Then, we assessed expression at the gene transcription level in freshly isolated keratinocytes. Then, trying to mimick alterations of the ECM of psoriatic keratinocytes we chose fibronectin (FN) of which over-expression reflects the most well described and characterized change in psoriasis (non-lesional skin). Therefore, we used FN as a substrate for psoriatic keratinocytes and compared it with healthy keratinocytes (Figure 3). The first 2 experiments corresponding to Figure 3a and b were done to confirm our experimental system ie that when FN is used as a substrate for cell culture in vitro, different responses are induced between healthy and psoriatic keratinocytes. For this reason, we assayed cell proliferation and TNFα as a marker for inflammation. Since we showed differences in the responses between two groups of keratinocytes we wanted to find out if syndecans’ expression is altered by the presence of FN. Finally, to enhance our data we assayed the effect of cytokine blockers, such as anti-TNFa and anti-IL17a (commonly used for the treatment of psoriasis) on the expression or detection of syndecans when delivered topically onto the epidermis. This experiment had a dual significance. It is known that proteoglycans including syndecans modulate the effect of cytokines, and cytokines can influence the expression of proteoglycans. Therefore, we thought relevant to see if syndecans are affected when these two cytokines are blocked as in the treatments for psoriasis and what effect this might have on the quality of the epidermis. From one other aspect, it is documented that biologics, such as anti-TNFa used to treat several auto-immune diseases and are routinely administered systematically may affect several organs likely by interefering non-specifically with signaling pathways in different sites of the organism. Thus, applying these drugs in a simpler system such as a skin equivalent would allow us to have more targeted results.

Point 3:  Why the Authors decided to check fibronectin effect on syndecans or to mix NHK with PHK in experiment showed on fig.3c. Moreover, rationale for inhibition of TNFa nad IL7a is scarce.

Response 3: Fibronectin and its receptor integrin a5b1 are well-characterized alterations in the basement membrane of non-lesional psoriatic skin. Moreover, fibronectin has heparan-binding domain through which it interacts with syndecans in different physiological conditions such as in wound healing. On top of it, in a previous paper Maata et al 1999 (included in the refereces of this manuscript) describe a mechanism of syndecan-1 regulation through a FiRE element (syndecan-1 gene enhancer) which is activated by the combination of KGF and fibronectin in migrating keratinocytes in wound healing. Finally, syndecans cooperate with integrins for the induction of signaling. For all these reasons, we considered fibronectin in this study.

Considering the co-culture of healthy keratinocytes with psoriatic fibroblasts it was selected based on previous published data reporting that when healthy keratinocytes and vice versa are co-cultured or are combined in 3d skin equivalent models the psoriatic phenotype can be reproduced (eg, formation of epidermis with hyperkeratosis n skin equivalents or alterations in the genetic programs of both cell types). Additionally, since fibronectin appears mainly in the non-lesional skin in psoriasis we thought that using healthy keratinocytes co-cultured with psoriatic fibroblasts we might reproduce the pre-activated state of keratinocytes of the non-lesional epidermis. We would have liked to have data for both ie using psoriatic keratinocytes in co-culture with psoriatic fibroblasts but psoriatic keratinocytes are hard to maintain in culture for a long time and we did not have enough at the time of this experiment.

Point 4: Why the authors applied MTT test for proliferation of cells analysis. The results are not reliable. Even in the present paper the results obtained with MTT were inconsistent with PCNA expression. The better method is BrdU or other tests.

Response 4: We thank the reviewer for this comment/suggestion. We agree that it might have been better to use another method or to confirm the MTT data with another method assaying cell proliferation. MTT is an assay used often in our lab and it gives some information about the metabolic activity of cells as an indication of biocomatibility or cell proliferation rate. In this study, MTT was used to evaluate and compare the proliferation rates of healthy versus psoriatic keratinocytes. In fact, observing psoriatic keratinocytes to the optic miscroscope at different time points it was obsious that the proliferation rate and the differentiation process were withheld much faster and quite differently than in healthy keratinocytes. Usually, keratinocytes in cell culture start to differentiate when they are in contact with other cells which is also an indication of the culture confluency. Nevertheless, we made the observation that from one day to another psoriatic keratinocytes in cell culture dishes presented alternative cycles were the cells were more or less confluent independent on the day of culture instead of increasing progressively and differentiate until they enter apoptosis. We assume that this “phenomenon” might have interfered with our MTT results. Contrary to that the data obtained by PCR analysis are more normalized since we quantify and use the same quantities of RNA in all conditions.

What we found interesting was that psoriatic keratinocytes seeded onto fibronectin appeared to grow faster with MTT analysis suggesting that there is some kind of regulation of the cell proliferation and/or differentiation process. A hypothesis might be that the transition of non-lesional pre-activated psoriatic keratinocytes to the hyperproliferative stage is finely tuned and the ECM composition plays a role in this process. Besides, fibronectin is found within the non-lesional epidermis and absent from the lesional basement membrane were keratinocytes are in a pre-active state but they are not hyperproliferative yet.

It is obvious though that the data from PCNA analysis may be confusing and since the number of replicates in only 3 as compared to the number in the MTT test and the result is not significant we decided to remove it from the graph.

Point 5:  Conclusion in the abstract suggesting that syndecans play a role in psoriasis and could be the target for therapy is not supported by the data.

Response 5: We agree with the reviewer. The sentence concluding that syndecans could serve as targets for therapy in the abstract was rephrased and also removed from the title.

Point 6:  Mechanistic aspect of this study is scarce

Response 6: In this study, the authors primarily wanted to answer the question if syndecans expressed by keratinocytes present any alteration in the lesional psoriatic skin implying its participation in the development of the disease. The data from experiments with FN may allow us to make the hypothesis of its involvement in the regulation of syndecan-1 in psoriasis. Nevertheless, we only made a hypothesis and we do not suggest that we elucidated the mechanism. Similarly, both syndecans are regulated by pro-inflammatory cytokines such as TNFa in several pathological and physiological conditions in different cell types. Additionally, pro-inflammatory cytokines can be bound to the heparan sulfate chains of syndecans and be finely regulated. Thus, we thought that it would be relevant to examine if pro-inflammatory cytokines inhibitors used as treatments for psoriasis directed immediately to the skin can affect syndecans’ expression. Again with this experiment we were trying to collect more data reinforcing our hypothesis that syndecans may play a role in the development of psoriasis but we are far from proposing a mechanism.

Point 7:  The anatomical localization of the samples collection is not given.

Response 7: The anatomical localization of the samples collections is added in the relevant section of materials and methods.

Point 8: The typing mistakes should be corrected

Response 8: The entire manuscript was re-read and typing errors were corrected.

Reviewer 2 Report

1. line 46 and throughout: " ... [2][3] ..." --> ... [2,3] ...

2. IL-17a , TNFa etc a --> α or alfa

3. line 62: " ... ECM ..." --> extracellular matrix (ECM)

4 line 72: " ... in vivo ... " Italicize this.

5. line 86: " ... GFs ... " Spell this out.

6. "Figure 1a" --> Figure 1 " Table 1 (Figure 1b)." --> Table 1

The legend to Figure 1 should be immediately below the Figure.

7. line 376: Indicate the age range of the healthy subjects.

8."Student's t-test (unpaired, two-tailed) was used for two-group comparisons."  For comparisons among more than two groups ANOVA with appropriate post-test should be used.

Author Response

Response to Reviewer 2 Comments

We thank the reviewer for taking the time to read and evaluate our paper. We addressed all comments and revised the manuscript accordingly.

Point 1: line 46 and throughout: " ... [2][3] ..." --> ... [2,3] ...

Response 1: The format of the references was adjusted accordingly in the text

Point 2: IL-17a , TNFa etc a --> α or alfa

Response 2: The correction was performed accroding to the suggestion

Point 3: line 62: " ... ECM ..." --> extracellular matrix (ECM)

Response 3: The correction was included in the text

Point 4: line 72: " ... in vivo ... " Italicize this.

Response 4: The correction was included in the text

Point 5 : " ... GFs ... " Spell this out.

Response 5: The abbreviation “GFs” is spelt out in a previous line (60).

Point 6: "Figure 1a" --> Figure 1 " Table 1 (Figure 1b)." --> Table

Response 6: The figure was separated as suggested and reference within the text was adjusted to that change.

Point 7: line 376: Indicate the age range of the healthy subjects.

Response 7: The age of healthy subjects was added.

Point 8: "Student's t-test (unpaired, two-tailed) was used for two-group comparisons."  For comparisons among more than two groups ANOVA with appropriate post-test should be used.

Response 8: For comparisons between more than two groups multiple t-tests were performed initially using the Prism GraphPad but it was not mentioned in the text by mistake. It is now added and detailed in the materials and methods section. However, if the reviewer insist we could perform ANOVA test.

Round 2

Reviewer 1 Report

Thank you very much for the correction of the manuscript made by the Authors.

I would like to recommend  additional small correction:

1. The limitations of the MTT test for measurement of the cell proliferation should be given in the text of the paper.

Author Response

Comment 1. The limitations of the MTT test for measurement of the cell proliferation should be given in the text of the paper.

Response 1: We thank the reviewer for this comment. We commended on the MTT limitations in the relative section (Section 2.2) in the text accompanied with some information from the literature.